# The Inhibitory Effect of *Hedera helix* and Coptidis Rhizome Mixture in the Pathogenesis of Laryngopharyngeal Reflux: Cleavage of E-Cadherin in Acid-Exposed Primary Human Pharyngeal Epithelial Cells

**DOI:** 10.3390/ijms252212244

**Published:** 2024-11-14

**Authors:** Nu-Ri Im, Byoungjae Kim, You Yeon Chung, Kwang-Yoon Jung, Yeon Soo Kim, Seung-Kuk Baek

**Affiliations:** 1Department of Efficacy Evaluation and Diagnosis Team, Zymedi, Incheon 21983, Republic of Korea; 2Department of Otorhinolaryngology-Head and Neck Surgery, College of Medicine, Korea University, Seoul 02841, Republic of Korea; 3Neuroscience Research Institute, College of Medicine, Korea University, Seoul 02841, Republic of Korea

**Keywords:** hedera and coptidis, laryngopharyngeal reflux, adenosine A3, matrix metalloproteinase 7, cadherins

## Abstract

Laryngopharyngeal reflux disease (LPRD) is a prevalent upper airway disorder characterized by inflammation and epithelial damage due to the backflow of gastric contents. Current treatments, primarily proton pump inhibitors (PPIs), often show variable efficacy, necessitating the exploration of alternative or adjunctive therapies. This study investigates the therapeutic potential of a mixture of Hedera helix and Coptidis rhizome (HHCR) in mitigating the pathophysiological mechanisms of LPRD. Using an in vitro model of human pharyngeal epithelial cells exposed to acidic conditions, we observed that acid exposure significantly increased the expression of adenosine A3 receptor (adenosine A3) and matrix metalloproteinase-7 (MMP-7), leading to E-cadherin cleavage and compromised epithelial integrity. Treatment with the HHCR mixture effectively suppressed adenosine A3 expression and MMP-7 activity, thereby reducing E-cadherin cleavage and preserving cellular cohesion. These results highlight the HHCR mixture’s ability to modulate the adenosine A3–MMP-7–E-cadherin pathway, suggesting its potential as a valuable adjunctive therapy for LPRD, particularly for patients unresponsive to conventional PPI treatment. This study provides new insights into the molecular interactions involved in LPRD and supports further clinical evaluation of HHCR as a complementary treatment option.

## 1. Introduction

The backflow of food materials, gastric acid, and pepsin from the stomach through the esophagus to the upper aerodigestive tract causes laryngopharyngeal reflux disease (LPRD) [1]. LPRD, which is characterized by persistent coughing, globus sensation, throat clearing, and hoarseness, is a prevalent and significant upper airway inflammatory disorder [2,3]. Currently, LPRD treatment involves dietary and lifestyle modifications as well as medications such as proton pump inhibitors (PPIs), which are the first-choice drugs for LPRD [4,5].

Evidence from randomized trials concluding that PPI treatment is superior to placebo remains insufficient. Moreover, clinical responses to these treatments and improvements in symptoms are relatively poor [4,6,7], and the inconsistency in disease or treatment responses in LPRD cannot be adequately explained using the currently proposed pathophysiology. Consequently, alternative therapeutic approaches beyond PPI therapy are urgently needed to manage LPRD.

Medicinal herbal remedies have played an important role in treatment from antiquity to the present day, and the use of medicinal plants as natural products for the production of semisynthetic derivatives has recently gained interest [8]. *Hedera helix* (common ivy, English ivy, European ivy, or ivy) is a European folk medicine that has been used for centuries to treat various diseases [9]. Furthermore, *H. helix* exerts therapeutic effects against upper respiratory tract infections. Notably, a clinical trial involving 9657 patients conducted by Fazio et al. in 2009 demonstrated anti-inflammatory, anti-congestive, antibacterial, and anti-spasmodic effects against acute and chronic bronchitis [10].

Coptidis rhizoma (CR) is the dried root stem of *Coptis chinensis* Franc, a member of the Ranunculaceae family, which has been used in China since ancient times. Recent studies have demonstrated effective suppression of airway inflammation, reduction in sputum production, and cough-relieving effects of an herbal mixture containing CR in animal models of citric acid-induced cough. Notably, significant antitussive effects have been reported using CR monotherapy [11].

AG NPP709 (Synatura^®^) represents a successful drug developed from natural products by Ahn-Gook Pharmaceuticals Co., Ltd. (Seoul, Republic of Korea). This drug contained a mixture of a 30% ethanolic extract of *H. helix* L. (ivy; Lamiaceae) leaves and a dried water-saturated butanolic extract of *C. chinensis* Franch (3:1, *w*/*w*). In 2011, the Korean Ministry of Food and Drug Safety approved AG NPP709 for the treatment of respiratory inflammatory disorders, and it has since become commercially available. AG NPP709 is effective for treating respiratory and chronic obstructive pulmonary diseases [9,12].

In a previous study, we reported an in vitro model of LPRD in which E-cadherin was degraded in human-derived pharyngeal mucosa exposed to acid [13]. This study aimed to investigate the effects of a mixture of *H. helix* and Coptidis rhizomes (HHCR) on the pathophysiological mechanisms of LPRD. To achieve this, we utilized an in vitro model of human pharyngeal epithelial cells exposed to an acidic environment to mimic LPRD conditions. This approach allows for the examination of cellular and molecular responses under controlled conditions and provides insights into the protective mechanisms of the HHCR mixture.

## 2. Results

### 2.1. Elevated Expression of Adenosine A3 by Acid Exposure in Pharyngeal Mucosal Epithelial Cells

A Korean pharmaceutical company (Ahn-Gook Pharmaceuticals Co., Ltd.) commissioned MDS Pharma Services, a global research institute, to elucidate the mechanism of action of the antitussive-expectorant activity of AG NPP709. The investigation revealed that AG NPP709 effectively inhibits tachykinin NK1, NK2, phosphodiesterase 4B (PDE4B), and adenosine A3 receptors, which are implicated in airway smooth muscle contraction, a key factor in cough pathogenesis. In addition, AG NPP709 effectively inhibited leukotriene B4 (LTB4), a major inflammatory mediator.

Based on our previous studies [13], we exposed the cells to an acidic medium (pH 4.0) for 0, 1, and 5 min and harvested them after 24 h. Reverse transcription PCR (RT-PCR) was performed for the tachykinins NK1, NK2, adenosine A3, PDE4B, and LTB4. Notably, adenosine A3 transcription significantly increased in a time-dependent manner after acid exposure. However, no significant changes were observed in the other genes (*p* < 0.05) (Figure 1A).

Cultured cells were treated with an adenosine A3 receptor antagonist to further investigate the effect of adenosine A3 in acidic environments, and cAMP levels were measured to evaluate its activity. Notably, adenosine A3 inhibition decreased adenyl cyclase activity and reduced cAMP levels [14]. As expected, cAMP levels decreased because of the increased adenosine A3 expression following acid exposure. Furthermore, no changes in cAMP levels were observed compared to the control when the antagonist was applied, indicating the effectiveness of the antagonist (Figure 1B). Moreover, treatment with an adenosine A3 antagonist significantly reduced the expression of adenosine A3, which was increased upon acid exposure (Figure 1C).

### 2.2. Adenosine A3 Induces E-Cadherin Cleavage Through MMP-7 Following Acid Exposure

Our previous study has demonstrated that MMP-7 induces E-cadherin cleavage in LPRD [13]. Additionally, another study reported that adenosine A3 reduces E-cadherin expression in head and neck squamous cell carcinoma, which is associated with epithelial–mesenchymal transition and contributes to changes in migration and invasion [15]. Therefore, we investigated the role of adenosine A3 in E-cadherin cleavage during acid exposure.

First, the cells were treated with an adenosine A3 antagonist and intracellular MMP-7 levels were measured to determine the relationship between adenosine A3 and MMP-7. The intracellular protein levels of the active form of MMP-7 decreased as the duration of acid exposure increased, whereas the levels in the culture media increased. However, no change in MMP-7 levels was observed when the cells were treated with either the MMP inhibitor or adenosine A3 antagonist (Figure 2A). The enzymatic activity of MMP-7 was analyzed to evaluate its activity. MMP-7 activity increased only in the acid-treated cells and was inhibited by MMP inhibitors. Additionally, the activity of MMP-7 decreased significantly after treatment with an adenosine A3 antagonist (Figure 2B). E-cadherin expression in adenosine A3 antagonist-treated cells following acid exposure was evaluated to confirm whether adenosine A3 plays an important role in E-cadherin cleavage. Notably, no changes were observed in E-cadherin or soluble form (sE-cadherin) levels in adenosine A3 antagonist-treated cells following acid exposure (Figure 2C). Furthermore, the inhibitory effect of adenosine A3 antagonist on MMP-7 and E-cadherin cleavage induced by acid exposure was confirmed by immunocytochemical staining (Figure 2D).

No changes in adenosine A3 levels were observed after MMP-7 treatment or intercellular E-cadherin cleavage (Appendix A). This observation suggests that according to the increase in acid-induced MMP-7 expression and the E-cadherin cleavage pathway proposed in our previous study [13], alterations in adenosine A3 expression, along with an increase in extracellular MMP-7 expression, are anticipated to occur following acid exposure. Therefore, adenosine A3 promotes MMP-7-induced E-cadherin cleavage triggered by acid exposure.

### 2.3. Inhibitory Effect of HHCR Mixture on E-Cadherin Cleavage Following Acid Exposure in Pharyngeal Epithelial Cells

We assessed the inhibitory effect of the HHCR mixture on E-cadherin cleavage following acid exposure. Adenosine A3 transcription was investigated based on the mechanism presented in Figure 2. Notably, the mRNA and protein levels of adenosine A3 significantly reduced following treatment with the HHCR mixture (*p* < 0.05; Figure 3A,B). Immunofluorescence staining revealed a decrease in adenosine A3 fluorescence (green) following treatment with the HHCR mixture (Figure 3C). The HHCR mixture-treated cells exhibited no changes in intracellular or extracellular MMP-7 protein levels (Figure 3D). However, the HHCR mixture significantly reduced MMP-7 enzyme activity (Figure 3E). Furthermore, no decrease in the cellular E-cadherin or increase in the sE-cadherin levels was observed in the culture media from the HHCR mixture-treated cells following treatment with the HHCR mixture (Figure 3F). Overall, these results indicate that the HHCR mixture inhibited E-cadherin cleavage by inhibiting adenosine A3 expression, which is an upstream mechanism in acidic environments.

## 3. Discussion

To date, several controversies persist regarding the diagnostic and therapeutic management of LPRD [6,16]. Notably, pH monitoring or multichannel intraluminal impedance–pH monitoring, which directly measures decreases in pH and reflux in the pharynx, is an objective diagnostic method for LPRD. However, this method is both invasive and expensive. Consequently, in most clinical settings, LPRD is diagnosed based on patient symptoms and laryngoscopic findings, followed by empirical treatment with PPIs. The success rate of PPI therapy varies widely, ranging from 18% to 87%, because of subjective diagnosis and empirical treatment [17]. Furthermore, a recent systematic review suggested that the nonresponse rate in empirical treatment trials based on PPIs may be as high as 40% of patients [18]. Therefore, management strategies have focused on other adjuvant drugs, including H2-receptor blockers, prokinetics, and antacids, as well as lifestyle modifications [5].

LPRD symptoms manifest in different forms. Therefore, administering an antitussive expectorant along with PPIs can help alleviate upper respiratory symptoms caused by LPRD [19]. Notably, the HHCR mixture, which has been proven effective in treating respiratory inflammatory diseases, is also commonly used [9]. Our experimental results indicate that the HHCR mixture exhibits a multifaceted mechanism of action against upper airway inflammation. It effectively inhibits several key inflammatory mediators, including tachykinins NK1, NK2, PDE4B, and adenosine A3, which induce airway smooth muscle contraction and cause coughing. The HHCR mixture effectively inhibited the activity of LTB4, a potent pro-inflammatory mediator.

Adenosine A3 is involved in modulating inflammatory responses and is linked to various diseases, including cardiovascular disease and cancer [14]. The activation of adenosine A3 can trigger pathways such as MAPK/ERK and NF-κB, leading to the upregulation of MMP-7. MMP-7 degrades extracellular matrix components, which contributes to tissue remodeling and cancer spread. The adenosine A3-MMP-7 axis is crucial for understanding disease mechanisms and may provide therapeutic options for inflammation and cancer treatment [20].

Our study focused on the role of adenosine A3 in LPRD. We observed a marked upregulation of adenosine A3 expression in pharyngeal mucosal epithelial cells following acid exposure. This increase in adenosine A3 expression induced E-cadherin cleavage via MMP-7. Furthermore, adenosine A3 expression increased following acid exposure and was inhibited by the HHCR mixture.

Our findings provide new insights into the therapeutic potential of the HHCR mixture, which is currently used as an adjunct to PPI therapy to manage the upper airway inflammation associated with LPRD. The HHCR mixture may exert beneficial effects, at least in part, by modulating the MMP-7 and E-cadherin pathways, which were implicated in LPRD pathogenesis in our previous studies [13,20].

In conclusion, this study provides comprehensive evidence that the cleavage of E-cadherin, a crucial cell–cell adhesion molecule induced under acidic conditions, is driven by the upregulation of adenosine A3 receptor and MMP-7 activity. Our findings revealed that treatment with the HHCR mixture effectively inhibited the acid-induced expression of adenosine A3 and subsequent MMP-7 activation, thereby reducing E-cadherin cleavage and preserving epithelial integrity. This aligns directly with the results obtained, offering new insights into the molecular pathways underlying LPRD and substantiating the therapeutic efficacy of the HHCR mixture. The demonstrated ability of this natural compound to modulate the adenosine A3–MMP-7–E-cadherin axis highlights its potential as a valuable adjunctive treatment for LPRD, especially in patients who show limited response to conventional PPI therapy. Future clinical studies are warranted to confirm these findings and explore the broader application of HHCR in managing LPRD.

## 4. Materials and Methods

### 4.1. Tissue Collection and Processing

Normal pharyngeal mucosa was harvested from the posterior pillar region of 21 volunteers (10 men and 11 women, aged 19–64 years) who underwent tonsillectomy for tonsillar hypertrophy and sleep-related disorders. The exclusion criteria were acute pharyngeal inflammation, history of allergies, smoking, and current pharmacological treatment. The study protocols and experiments were reviewed and approved by the Institutional Review Board of the Korea University Hospital (IRB no. ED15303). All participants provided signed informed consent, and all methods were performed in accordance with the relevant guidelines and regulations.

The collected samples were stored in 1Xpbs to contain the tissue, and cell culture was achieved within 1 h.

### 4.2. Cell Culture

Human pharyngeal mucosal samples were incubated in 1 mg/mL dispase in Dulbecco’s modified Eagle’s medium/F12 (DMEM/F12) for 1 h at 37 °C and 5% CO_2_. The cells were rinsed thrice in DMEM/F12 and subsequently cultured in a bronchial epithelial growth medium (BEBM; Lonza, Walkersville, MD, USA) [21]. All experiments were performed using pharyngeal epithelial cells from the second passage onwards. Routine cultures were maintained in a 5% CO_2_ incubator at 37 °C, and the medium was changed every 3 days. The cell morphology was examined using an Olympus CKX41-A32PHP microscope (Olympus, Tokyo, Japan).

### 4.3. Treatment with Acid, Adenosine A3 Receptor (Adenosine A3) Antagonist, MMP Inhibitor, and HHCR Mixture

To mimic acid reflux, confluent pharyngeal epithelial cells were treated with HCl (pH 4.0) for 1 or 5 min. The cells were then incubated with a non-acidic BEBM medium at 37 °C and 5% CO_2_ for 24 h, washed twice with PBS, and used for the experiments.

Epithelial cells were pretreated with a 5 nM adenosine A3 antagonist (Tocris, Bristol, UK) and incubated for 1 h at 37 °C and 5% CO_2_. After treatment with HCl, the cells were incubated with 5 nM adenosine A3 antagonist for 24 h. Control cells were treated with non-acidic BEBM and 5 nM adenosine A3 antagonist for the same period.

Epithelial cells were pretreated with 10 μM actinonin (MMP inhibitor, Santa Cruz Biotechnology, Santa Cruz, CA, USA) for 1 h at 37 °C and 5% CO_2_ to investigate the effect of MMP inhibition. After treatment with HCl as described above, cells were incubated with 10 μM actinonin for 24 h. Control cells were treated with non-acidic BEBM and 10 μM actinonin for the same period.

Ivy leaf dry extracts of H. helix and CR were purchased from Ahn-Gook Pharmaceutical Co., Ltd. The HHCR mixture was prepared by dissolving the herbal extracts of H. helix and CR in PBS at a volume ratio of 3:1. Epithelial cells were pretreated with 8 µL of HHCR mixture for 10 min at 37 °C and 5% CO_2_ to investigate the effects of the HHCR mixture. After treatment with HCl, as described above, the cells were incubated in BEBM for 24 h. Control cells were treated with a non-HHCR mixture BEBM for the same period.

### 4.4. Real-Time Polymerase Chain Reaction (PCR)

Gene expression in the epithelial cells was determined using qPCR. Total RNA was extracted from approximately 5 × 10^5^ cells using the TRIzol reagent (Qiagen, Hilden, Germany) and RNase-free DNase I (Qiagen). The 1 ug RNA was reverse-transcribed into cDNA using AmfiRivert cDNA synthesis platinum master mix (GenDEPOT, Katy, TX, USA). The prepared cDNA was amplified and quantified using SYBR green master mix (Qiagen) using the following primers: GAPDH, forward 5′-GAG TCA ACG GAT TTG GTC GT-3′ and reverse 5′-TTG ATT TTG GAG GGA TCT CG-3′; Tachykinin NK 1, forward 5′-GA CTG TGC TGA TCT ACT TCC-3′ and reverse 5′-AG CTC TCT GTC ATG GTC TTG-3′; Tachykinin NK 2, forward 5′-TT ATT GCT GGC ATC TGG CTG-3′ and reverse 5′-GA GCT TAT CTT CCT TGG TGG-3′; adenosine A3 forward 5′-GG CTG CCC TCA AAT AAC ATC-3′ and reverse 5′-CT CCA CCT CTT CAC TTC TG-3′; PDE 4B forward 5′-AT TCT GTT TGT CCA GGA ATG-3′ and reverse 5′-AT GCT GGT GTA GAA AGG AGA-3′; LTB4 forward 5′-GA GTT CAT CTC TCT GCT GGC-3′ and reverse 5′-CC AGG TTC AGC ACC ATC AGG-3′. The PCRs were performed using a real-time thermal cycler system (TP800/TP860, Takara, Kusatsu, Shiga, Japan) with 40 cycles of a 2-step reaction involving denaturation at 95 °C for 15 s, followed by annealing/extension at 60 °C for 45 s. Data were analyzed using the △Ct method.

### 4.5. Cyclic AMP Analysis

The cAMP levels in the supernatant of cells treated with the acid or acid/adenosine A3 antagonist were quantified to confirm the inhibitory effect of the adenosine A3 antagonist using an ELISA kit, following the manufacturer’s instructions (catalog# EMSCAMPL; Thermo Fisher Scientific, Waltham, MA, USA).

### 4.6. Western Blotting

Human pharyngeal epithelial cells treated with acid, an acid/MMP inhibitor, an acid/adenosine A3 antagonist, or an acid/HHCR mixture were collected by scraping to maintain cell–cell interactions without single-cell dissociation. The supernatant of each well was concentrated to equal volume using Centricon (3 kDa cutoff; Merck Millipore, Billerica, MA, USA) at 3000× *g* for 40 min at 4 °C. Each sample was then mixed with 5× Laemmli buffer and 5% β-mercaptoethanol and boiled for 10 min. Briefly, equal volumes of protein were separated using sodium dodecyl sulfate–polyacrylamide gel electrophoresis and transferred onto nitrocellulose membranes. The membranes were incubated with E-cadherin (1:1000; Santa Cruz Biotechnology), MMP-7 (1:1000; Sigma-Aldrich, St. Louis, MO, USA), or adenosine A3 (1:1000; Abcam, Cambridge, UK) antibodies overnight at 4 °C, or β-actin antibody (1:2000; Santa Cruz Biotechnology) for loading control in blocking solution. Next, the membranes were incubated with appropriate anti-rabbit (1:1000; Santa Cruz Biotechnology) or anti-mouse (1:1000; Santa Cruz Biotechnology) antibodies in a blocking solution. The blots were visualized using a chemiluminescence kit (Santa Cruz Biotechnology), and images were captured using ChemiDoc (Bio-Rad Laboratories, Hercules, CA, USA).

Preserving cell–cell junction adhesions is crucial for evaluating E-cadherin expression. Therefore, the difference in E-cadherin expression was semi-quantitatively estimated based on β-actin expression in each sample instead of quantitative protein analysis after single-cell dissociation. The amount of protein in the supernatant of each sample was determined using a bicinchoninic acid protein assay, with bovine serum albumin as the standard, and equal amounts of protein were loaded onto the gel.

### 4.7. MMP-7 Enzyme Activity Measurement

Culture media from pharyngeal epithelial cells treated with acid, an acid/MMP inhibitor, an acid/adenosine A3 antagonist, or an acid/HHCR mixture were used to measure MMP-7 activity. The concentrate was incubated with 10 μM 4-aminophenylmercuric acetate in assay buffer for 1 h at 37 °C. After adding diluted MMP-7 substrate (5-FAM/QXLTM 520 FRET peptide; SensoLyte 520 MMP-7 assay kit, Anaspec Inc., Fremont, CA, USA) at a 1:100 ratio in assay buffer and incubating for 1 h at 37 °C, the enzyme activity was determined at excitation and emission wavelengths of 490 and 520 nm, respectively, using the SOFTMAX PRO v5 software (Molecular Devices, Sunnyvale, CA, USA) with a SpectraMax M2e plate reader (Molecular Devices).

### 4.8. Immunocytochemistry Analysis

Cells were cultured on cytoslides (Marienfeld-Superior, Lauda-Königshofen, Germany) and treated with acid and acid/adenosine A3 antagonists as mentioned above. The cells were fixed with 4% glutaraldehyde for 30 min and blocked with goat serum (Vector Laboratories, Burlingame, CA, USA) for 1 h at room temperature. The cells were then incubated with rabbit polyclonal antibody against E-cadherin (1:100; Santa Cruz Biotechnology) or MMP-7 (1:200; Sigma) overnight at 4 °C. The following day, cells were treated with a biotinylated anti-rabbit IgG (H + L) secondary antibody in PBS (1:400) for 60 min at room temperature. After washing with PBS, antigen–antibody complexes were detected using an avidin–biotin complex detection system (Vectastain ABC Kit; Vector Laboratories). The cytoslides were stained using a DAB Substrate Kit (Vector Laboratories), rinsed in water, counterstained with Mayer’s hematoxylin, and washed again in water. After mounting on glass slides, the slides were examined using an Axio Scan slide scanner (Z1; ZEISS, Oberkochen, Germany).

### 4.9. Immunofluorescence Analysis

Cells were cultured on cytoslides and treated with an acid or an acid/HHCR mixture as described above. Cells were fixed with 4% formaldehyde for 30 min and blocked with goat serum (Vector Laboratories) for 1 h at room temperature. After washing in PBS, the slides were incubated overnight at 4 °C with adenosine A3 antibody (1:200, Abcam, Cambridge, UK) and then treated with Alexa Fluor^®^ 488 goat anti-rabbit secondary antibody (Abcam) for 2 h, and the nuclei were counterstained with 4′,6-diamidino-2-phenylindole (DAPI; Sigma-Aldrich) at a concentration of 0.1 μg/mL in PBS. Adenosine A3 and DAPI presented green and blue fluorescence, respectively, and images were captured using an Axio Scan slide scanner (ZEISS).

Immunostaining for E-cadherin, MMP-7, and adenosine A3 was performed in five microscopic fields (200× magnification) in three different samples. The semi-quantitative score was calculated as the percentage of stained cells per total number of cells in each microscopic field. The results were independently evaluated by three researchers.

### 4.10. Statistical Analysis

The statistical results are denoted as mean value ± standard deviation (SD) measured after each experiment (N). The data were based on a one-way analysis of variance (ANOVA) and a *p*-value of less than 0.05 was considered statistically significant.

## Figures and Tables

**Figure 1 ijms-25-12244-f001:**
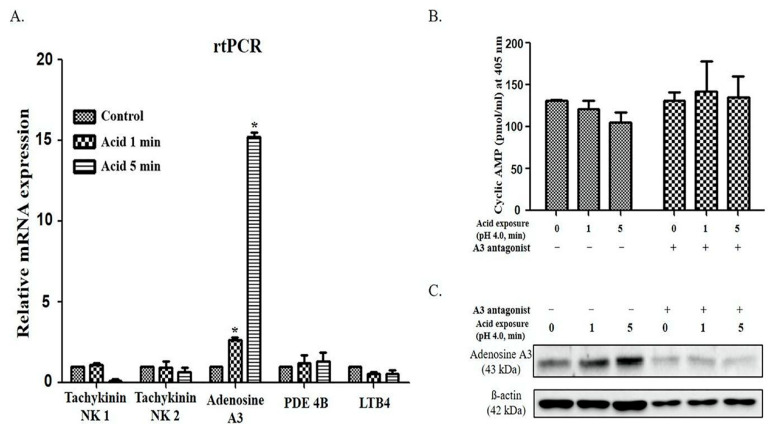
Changes in adenosine A3 expression following acid exposure: (**A**) qPCR analysis revealed an increase in the mRNA levels of adenosine A3 with the increase in the duration of acid exposure 1 or 5 min (*p* < 0.05); (**B**) ELISA revealed no changes in the cyclic AMP expression following treatment with the adenosine A3 receptor antagonist; (**C**) adenosine A3 expression increased with the increase in the duration of acid exposure time, whereas these changes were not observed following treatment with an adenosine A3 antagonist. Statistical analysis was performed using one-way ANOVA, and error bars represent [SD]. Asterisks indicate statistically significant differences (*p* < 0.05).

**Figure 2 ijms-25-12244-f002:**
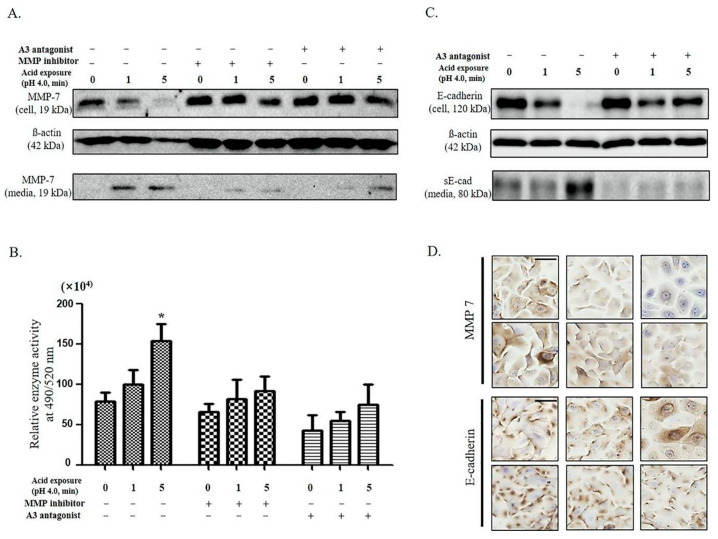
Role of adenosine A3 in E-cadherin cleavage following acid exposure: (**A**) decrease in the intracellular (cell) and increase in the extracellular (media) levels of the active form of MMP-7 depending on the duration of the acid exposure time. These changes were not observed following treatment with the adenosine A3 antagonist. (**B**) While MMP7 enzyme activity was statistically significantly increased in proportion to acid exposure time, it was confirmed that there was no change when treated with adenosine A3 antagonist. (**C**) The intracellular cleavage (cell) and extracellular secretion (media) of E-cadherin following acid exposure time. These changes were not observed following treatment with the adenosine A3 antagonist. (**D**) The immunocytochemical staining images show MMP-7 and E-cadherin expression. Scale bar = 200 µm. The lower row of panel D indicates no decrease in the intracellular levels of MMP-7 or E-cadherin cleavage following the treatment with the adenosine A3 antagonist. Statistical analysis was performed using one-way ANOVA, and error bars represent [SD]. Asterisks indicate statistically significant differences (*p* < 0.05).

**Figure 3 ijms-25-12244-f003:**
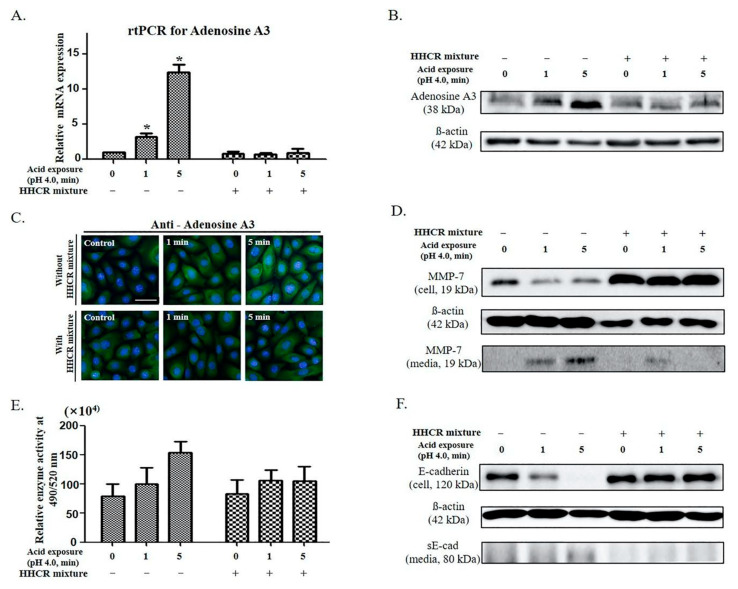
Effect of the treatment with the HHCR mixture on E-cadherin cleavage following acid exposure in human pharyngeal mucosal epithelial cells: (**A**) It was confirmed that adenosine A3, which increased statistically significantly in proportion to acid exposure time, was suppressed when treated with the HHCR mixture. (**B**) Adenosine A3 protein expression in epithelial cells was significantly suppressed following the treatment with the HHCR mixture. (**C**) Immunofluorescence images show the effect of treatment with adenosine A3 antagonist for 24 h with or without acid exposure (green: adenosine A3, blue: DAPI). Scale bar = 50 µm. (**D**) Low intracellular expression of MMP-7 and high extracellular expression of MMP-7 (media) after acid exposure were significantly reversed by the HHCR mixture. (**E**) Increased MMP-7 enzymatic activity in an acidic environment was inhibited by the HHCR mixture. (**F**) E-cadherin cleavage was inhibited in the cells treated with the HHCR mixture. Statistical analysis was performed using one-way ANOVA, and error bars represent [SD]. Asterisks indicate statistically significant differences (*p* < 0.05).

## Data Availability

Data are contained within the article or Appendix A.

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
