# Peer review of "The Inhibitory Effect of *Hedera helix* and Coptidis Rhizome Mixture in the Pathogenesis of Laryngopharyngeal Reflux: Cleavage of E-Cadherin in Acid-Exposed Primary Human Pharyngeal Epithelial Cells"

_ijms, 2024, doi:10.3390/ijms252212244_

Round 1

Reviewer 1 Report

Comments and Suggestions for Authors

Comments and Suggestions for Authors

Dear Authors,

It’s a pleasure allowing me to review the paper titled “ijms-3296838 Inhibitory Effect of Hedera helix and Coptidis Rhizome Mixture in the Pathogenesis of Laryngopharyngeal Reflux: Cleavage of E-Cadherin in Acid-exposed Primary Human Pharyngeal Epithelial Cells” submitted to the “Biochemistry” section. This paper addresses the therapeutic investigations of a mixture of two herbal natural products Hedera helix and Coptidis rhizome to treat laryngopharyngeal reflux disease.

The abstract comprehensively summarizes the content of the article.

The keywords should be reviewed and adjusted according to MeSH classification.

The introduction highlights the importance to treat the laryngopharyngeal reflux disease by reporting challenges of the existing therapy. the objective should be implemented adding what in vitro model was used for the study.

Results are well structured, making them easy to follow, and are supported by figures, but figure legends are not supported from the detailed description of the graphs, what the bar charts represent including the error bars, ecc., and explanation added into text.

The discussion assesses the results; however, some references should be added in lines 187, 188 and 193. The discussion is a little sparse, please provide implementation on the results obtained in other previous studies, if existing, confirming or not the results obtained.

Materials and methods

4.1 title “Tissue preparation” seems like “Tissue collection”, otherwise the authors are encouraged to add how the tissues were processed after collection.

4.3 The description of control cells to the treatment with acid is lacking. The description of control cells to the treatment with ivy leaf dry extracts of H. helix and CR is lacking.

4.10 The statistical analysis applied to results obtained is lacking, please provide the statistic tests that were applied to analyzed data.

Finally, a general conclusion should be added that directly aligned with the results obtained in this study.

Author Response

Dear Editor,  

Please find enclosed the revised version of manuscript entitled “Inhibitory Effect of Hedera helix and Coptidis Rhizome Mixture in the Pathogenesis of Laryngopharyngeal Reflux: Cleavage of E-Cadherin in Acid-exposed Primary Human Pharyngeal Epi-thelial Cells” and point by point response to the comments made by the reviewer.

We would like to thank the reviewers for their constructive criticism, which has helped us improve the manuscript. We have attempted to carefully and thoroughly address all the reviewer’s concerns.

Thank you again for your prompt handling and advice regarding our submission.

We hope the current version of the manuscript is suitable for publication in IJMS.

Point-by-Point Response to reviewer’s Comments

We appreciate the time and efforts by the editor and referees in reviewing this manuscript. We have addressed all issues indicated in the review report, and believed that the revised version can meet the journal publication requirements.  

Reviewer #1:

This is a good paper overall. I have a few, mostly minor comments and suggestions.

  1. The abstract comprehensively summarizes the content of the article.

I have refined the abstract to better reflect the content of the paper. Thank you for your guidance and feedback.

“Laryngopharyngeal reflux disease (LPRD) is a prevalent upper airway disorder characterized by inflammation and epithelial damage due to the backflow of gastric contents. Current treatments, primarily proton pump inhibitors (PPIs), often show variable efficacy, necessitating the exploration of alternative or adjunctive therapies. This study investigates the therapeutic potential of a mixture of Hedera helix and Coptidis rhizome (HHCR) in mitigating the pathophysiological mechanisms of LPRD. Using an in vitro model of human pharyngeal epithelial cells exposed to acidic conditions, we observed that acid exposure significantly increased the expression of adenosine A3 receptor (adenosine A3) and matrix metalloproteinase-7 (MMP-7), leading to E-cadherin cleavage and compromised epithelial integrity. Treatment with the HHCR mixture effectively suppressed adenosine A3 expression and MMP-7 activity, thereby reducing E-cadherin cleavage and preserving cellular cohesion. These results highlight the HHCR mixture’s ability to modulate the adenosine A3/MMP-7/E-cadherin pathway, suggesting its potential as a valuable adjunctive therapy for LPRD, particularly for patients unresponsive to conventional PPI treatment. This study provides new insights into the molecular interactions involved in LPRD and supports further clinical evaluation of HHCR as a complementary treatment option.”

  1. The keywords should be reviewed and adjusted according to MeSH classification.

Thank you for your valuable suggestion. We have revised the keywords according to the MeSH classification.

Hedera and Coptidis; laryngopharyngeal reflux; adenosine A3; Matrix Metalloproteinase 7; Cadherins

  1. The introduction highlights the importance to treat the laryngopharyngeal reflux disease by reporting challenges of the existing therapy. the objective should be implemented adding what in vitro model was used for the study.

Thank you for your valuable suggestion. We have incorporated the details about the in vitro model into the introduction as per your suggestion.

“To achieve this, we utilized an in vitro model of human pharyngeal epithelial cells ex-posed to an acidic environment to mimic LPRD conditions. This approach allows for the examination of cellular and molecular responses under controlled conditions and pro-vides insights into the protective mechanisms of the HHCR mixture.”

  1. Results are well structured, making them easy to follow, and are supported by figures, but figure legends are not supported from the detailed description of the graphs, what the bar charts represent including the error bars, ecc., and explanation added into text.

Thank you for your feedback. We have revised all figure legends to include detailed descriptions of the graphs, what the bar charts represent, and clarifications regarding the error bars, as per your suggestion. The additions have been highlighted in red for your review.

  1. The discussion assesses the results; however, some references should be added in lines 187, 188 and 193. The discussion is a little sparse, please provide implementation on the results obtained in other previous studies, if existing, confirming or not the results obtained.

We have added references to support the statements mentioned. Thank you for your attention to detail. Additionally, we have further enhanced the discussion section with the assistance of your valuable input.

  1. Materials and methods

4.1 title “Tissue preparation” seems like “Tissue collection”, otherwise the authors are encouraged to add how the tissues were processed after collection.

Thank you for your feedback. We have revised the title of section 4.1 from 'Tissue preparation' to 'Tissue collection' and added details about the processes that followed tissue collection. The subsequent procedures are described in the 'Cell culture' section.

4.3 The description of control cells to the treatment with acid is lacking. The description of control cells to the treatment with ivy leaf dry extracts of H. helix and CR is lacking.

Thank you for your feedback. In response to your suggestion, we have added a description of the control cells in section 4.3 as follows:

'Control cells were treated with non-HHCR mixture BEBM for the same period.

4.10 The statistical analysis applied to results obtained is lacking, please provide the statistic tests that were applied to analyzed data.

Thank you for your feedback. We have addressed this by adding relevant information to the figure legends and section 4.10 of the Methods. These additions include descriptions of the statistical tests used, the definitions of asterisks indicating p-values, and the clarification of error bars.

“Statistical analysis was performed using one-way ANOVA, and error bars represent [SD]. Asterisks indicate statistically significant differences (p < 0.05).”

  1. Finally, a general conclusion should be added that directly aligned with the results obtained in this study.

 Thank you for your valuable feedback. We have supplemented the final part of the discussion with a more detailed conclusion that aligns with our experimental results.

Again, we appreciate the opportunity to revise our work for consideration for publication in Biomimetics. We hope our revision meet your approval. 

Reviewer 2 Report

Comments and Suggestions for Authors

See attached

Author Response

Please find enclosed the revised version of manuscript entitled “Inhibitory Effect of Hedera helix and Coptidis Rhizome Mixture in the Pathogenesis of Laryngopharyngeal Reflux: Cleavage of E-Cadherin in Acid-exposed Primary Human Pharyngeal Epi-thelial Cells” and point by point response to the comments made by the reviewer.

We would like to thank the reviewers for their constructive criticism, which has helped us improve the manuscript. We have attempted to carefully and thoroughly address all the reviewer’s concerns.

Thank you again for your prompt handling and advice regarding our submission.

We hope the current version of the manuscript is suitable for publication in IJMS.

Point-by-Point Response to reviewer’s Comments

We appreciate the time and efforts by the editor and referees in reviewing this manuscript. We have addressed all issues indicated in the review report, and believed that the revised version can meet the journal publication requirements.  

Reviewer #2:

  1. Use of internationally recognized gene and protein identifiers would be helpful to the reader to properly identify what molecules are under discussion, e.g. gene symbols designated by the HUGO Gene Nomenclature Committee (HGNC). Gene names should be in ALL CAPS and italicized and protein designations should be the same as the gene symbol, but not italicized.

Thank you for your valuable feedback. We appreciate your point, and we apologize for any confusion caused by the use of italics for protein names. We have revised the text to ensure that protein designations are not italicized

  1. Additional discussion of known adenosine A3 receptor biological functions and associated diseases (e.g. inflammation or cancer) and potential mechanisms by which it could lead to MMP-7 activation would provide more context to aid interpretation of the implications of the findings.

Your feedback was invaluable. We have incorporated a discussion on the relationship between the adenosine A3 receptor and MMP7.

“The adenosine A3 is involved in modulating inflammatory re-sponses and is linked to various diseases, including cardiovascular disease and cancer. Activation of adenosine A3 can trigger pathways such as MAPK/ERK and NF-κB, leading to the upregulation of MMP-7. MMP-7 degrades extracellular matrix components, which contributes to tissue remodeling and cancer spread. Theadenosine A3-MMP-7 axis is crucial for understanding disease mechanisms and may provide therapeutic options for inflammation and cancer treatment.”

  1. The statistical tests was not described in methods or figure legends. Asterisks (p=?) and error bars (SD/SE?) in figures were not defined.             

Thank you for your feedback. We have addressed this by adding relevant information to the figure legends and section 4.10 of the Methods. These additions include descriptions of the statistical tests used, the definitions of asterisks indicating p-values, and the clarification of error bars.

“Statistical analysis was performed using one-way ANOVA, and error bars represent [SD]. Asterisks indicate statistically significant differences (p < 0.05).”

Again, we appreciate the opportunity to revise our work for consideration for publication in Biomimetics. We hope our revision meet your approval. 
